# Prognostic Significance of BIRC5/Survivin in Breast Cancer: Results from Three Independent Cohorts

**DOI:** 10.3390/cancers13092209

**Published:** 2021-05-04

**Authors:** Nina Oparina, Malin C. Erlandsson, Anna Fäldt Beding, Toshima Parris, Khalil Helou, Per Karlsson, Zakaria Einbeigi, Maria I. Bokarewa

**Affiliations:** 1Department of Rheumatology and Inflammation Research, Institute of Medicine, University of Gothenburg; 40530 Gothenburg, Sweden; malin.erlandsson@rheuma.gu.se (M.C.E.); maria.bokarewa@rheuma.gu.se (M.I.B.); 2Rheumatology Clinic, Sahlgrenska University Hospital, 41345 Gothenburg, Sweden; 3Department of Medicine and Oncology, Southern Älvsborg Hospital, 50182 Borås, Sweden; anna.faldt.beding@vgregion.se (A.F.B.); zakaria.einbeigi@oncology.gu.se (Z.E.); 4Department of Oncology, Institute of Clinical Science at Sahlgrenska Academy, University of Gothenburg, 40530 Gothenburg, Sweden; toshima.parris@oncology.gu.se (T.P.); khalil.helou@oncology.gu.se (K.H.); per.karlsson@oncology.gu.se (P.K.); 5The King Gustav Vth Jubilee Clinic, Sahlgrenska University Hospital, 41345 Gothenburg, Sweden

**Keywords:** BIRC5, survivin, breast cancer, survival probability, molecular signature

## Abstract

**Simple Summary:**

Survivin, coded by the BIRC5 gene, is the cell death preventing protein, which is important for cell division in normal and cancer cells. It is intensively studied as a cancer biomarker and target for antitumor therapy. In this study we asked if we could get clinically helpful information on how active BIRC5 is in breast cancer patients? We studied the BIRC5 protein level in tumor samples for breast cancer patients from a West Swedish cohort and its mRNA level in two different public gene expression databases. Survival analysis demonstrated that a higher BIRC5 protein or mRNA level was associated with poor survival in all cohorts and for different cancer subtypes. We show that BIRC5 is a promising independent cancer survival marker.

**Abstract:**

Breast cancer (BC) histological and molecular classifications significantly improved the treatment strategy and prognosis. Inhibitor of apoptosis BIRC5/survivin is often overexpressed in cancers, however, indications of its importance in BC are inconsistent. We integrate BIRC5 protein and mRNA measures with clinical associates and long-term outcome in three independent cohorts Protein levels of BIRC5 were measured in primary lysates of 845 patients of the West Swedish BC cohort (VGR-BC) and linked to 5- and 27-years survival. The results were externally validated in transcriptomic data from METABRIC and SCAN-B cohorts. Survival analysis showed that high levels of BIRC5 were consistently associated with a poor probability of 5-year overall survival. High BIRC5 in VGR-BC contributed negatively to the disease-specific survival at 5 and 27 years. Subsets with different status by ER (estrogen receptor) expression and presence of nodal metastasis supported independent association of high BIRC5 with poor prognosis in all cohorts. In METABRIC and SCAN-B cohorts, high levels of BIRC5 mRNA were associated with the basal-like and luminal B molecular BC subtypes and with increasing histologic grade. BIRC5 is a sensitive survival marker that acts independent of ER and nodal status, and its levels need to be considered when making treatment decisions.

## 1. Introduction

Inspiring progress in the diagnosis and treatment of BC (breast cancer) patients was achieved. These advances were largely due to mammography screening programs introduced on a population-wide basis during 1980s, leading to early diagnosis. Nevertheless, several studies emphasized that treatment outcomes and survival prognosis of BC patients remains uneven [1,2]. This inconsistency is attributed to the variety of clinical and histopathological characteristics of BC. To overcome BC heterogeneity, integration of histological and genomic data are used for molecular classification of BC in clinical practice today. It also fuels development of new therapeutic approaches and contributes to expectations to improve the long-term prognosis for BC patients. Histological grading of BC by Scarff-Bloom-Richardson was introduced to measure a degree of deviation in BC from normal breast tissue through a tubule duct formation, mitotic activity, and nuclear pleomorphism [3] and was demonstrated to be superior in tumor size and the TNM (primary tumor size, involvement of lymph nodes, and status of distant metastasis) staging system, for BC prediction of major clinical outcomes, survival, and treatment response [4,5].

Molecular classification of BC by Perou and Sorlie proposed division of BC into four subgroups based on gene expression. It also reflects the distinct molecular mechanisms in BC development [6]. These subgroups are (a) luminal, with expression of estrogen receptor (ER), ER regulatory partners and genes, normally expressed in luminal cells; (b) the human epidermal growth factor receptor (HER)2 positive; (c) basal with no expression of hormonal receptors and HER2, and characterized by the expression of genes normally active in breast basal cells; and (d) normal-like carcinoma. The luminal BC subtype was separated further into different subtypes, the commonly ones accepted include the luminal A and B. For the luminal A, the hormonal receptors and related genes are shown to be expressed at high levels, while the luminal B subtype is recognized by the high expression of proliferation signature genes [7]. Although this classification is under constant improvement and takes advantage of modern developments in gene profiling, bioinformatic analysis, and treatment, the initial subtypes are generally recognized and used for clinical purposes [8,9].

The gene of Baculoviral Inhibitor of apoptosis Repeat Containing 5 (*BIRC5*) codes for protein survivin. The BIRC5/survivin protein was studied extensively during the past two decades [10]. The *BIRC5* gene was active during fetal development and renewal of normal healthy cells. It encodes multifunctional protein survivin, which plays different roles in a cell, depending on cellular localization. In the nucleus, survivin participates in the formation of the chromosome passenger complex and controls cell division, while cytoplasmic and mitochondrial survivin plays a pivotal role in inhibiting apoptosis. Survivin is involved in diverse molecular network of cancer-related processes, including tumor cell proliferation, invasive growth, and distant migration. BIRC5/survivin is highly expressed in many cancers, including BC [11]. It was highlighted as a promising prognostic biomarker and a potential oncological drug target. Recent bioinformatics evaluations conducted in independent datasets support the key role of survivin in BC [12,13]. However, the clinical significance as well as the molecular mechanisms behind BIRC5/survivin involvement in BC development remains unclear. It was proposed that BIRC5/survivin supports the aggressive tumor growth, leading to increased resistance to chemo- and radiotherapy. However, most of these studies were experimental and were based on the results obtained in immortalized cell lines or ex vivo tissue cultures [14,15]. BIRC5/survivin was recently evaluated for its potential use as a prognostic biomarker for BC [16,17,18]. Nuclear localization of BIRC5/survivin was demonstrated to be an independent beneficial factor for BC. High BIRC5/survivin levels often correlated with histological type [19,20], and positive expression of proliferative genes including Ki67 [21,22,23]. Recent meta-analysis revealed high BIRC5/survivin expression in BC to be predictive for disease-free and overall survival in hormone receptor negative BC [24]. However, a prospective Danish study was unable to establish a link between BIRC5/survivin expression ratio between the cytoplasm and nucleus, and the recurrence rate in a cohort of ER-positive BC patients [25]. Nevertheless, overexpression of cytoplasmic BIRC5/survivin was found to be associated with an increased risk of recurrence in ER-negative BC patients [24].

Treatment with BIRC5/survivin inhibitor YM155, sepantronium bromide, produced promising results and led to the regression of human triple negative BC metastases in mouse models [26]. A phase-II multicenter study using combination therapy of YM155 with the cytostatic drug docetaxel failed to show any significant difference in progression-free survival and other secondary endpoints, as compared to docetaxel alone [27]. Extracellular BIRC5/survivin is recognized by antigen presenting cells carrying HLA-A1, A2, and B35 receptors, and induced both cytotoxic lymphocyte response of CD8+ T cells, and survivin-specific antibody production [28,29]. These properties of BIRC5/survivin made it a subject of dendritic cell-based vaccination and anti-tumor sensitization protocols [30,31], which are to be tried out in clinical trials for patients with advanced solid tumors including BC, melanoma, glioma, and pancreatic and colon cancers.

In this study, we assess the clinical utility of the protein and mRNA BIRC5/survivin levels in primary BC tissue as an independent prognostic factor for overall and disease-specific survival in BC. Five-year survival analysis demonstrates that high BIRC5/survivin levels are associated with poor survival probability of BC patients in three independent cohorts. The analyses of the protein (VGR-BC cohort) and mRNA (METABRIC; Molecular Taxonomy of Breast Cancer International Consortium) and SCAN-B (Sweden Cancerome Analysis Network—Breast) cohorts) BIRC5/survivin levels produce consistent results that are independent of the ER and the nodal status of the patients. In the METABRIC and SCAN-B cohorts, high levels of BIRC5/survivin are associated with the basal-like and luminal B molecular BC subtypes, and with increasing histological grade.

## 2. Materials and Methods

### Patient Cohorts

The Jubilee Clinic Biobank at the Sahlgrenska University Hospital comprised a collection of 845 tissue samples with primary BC, here the VGR-BC cohort. All patients were diagnosed with BC in the West Götaland region of Sweden between years 1989–1995. The patients received the treatment of BC according to the common guidelines developed and updated by the Swedish Breast Cancer Group, www.swebcg.se (accessed on 12 February 2020) [32], which is generally in line with the semiannually updated international breast cancer St. Gallen consensus guidelines [33]. The clinical characteristics of the VGR-BC patients were extracted from the Swedish Cancer register and contained the ICD code of the malignancy and also information about the histological systematized nomenclature of medicine code, the TNM stage of the disease, the date of diagnosis, local and distant recurrence, and date of death.

All procedures were done in accordance with the Declaration of Helsinki and approved by the Medical Faculty Research Ethics Committee (Gothenburg, Sweden). Tumor material used in this study was extracted during the operation. The material was sent for the assessment of the hormone receptor status of the tumors and the tumor tissue, which was left after those analyses was collected from the laboratory of Clinical Chemistry. The Ethical permit was obtained to use the tumor tissues surgically extracted during operation for genetic and expression analysis (reference number 164-02 from 23 April 2002). The clinical information for the corresponding tumor material was extracted through the Swedish Cancer Register. Patients provided their informed consent for the collection of data required for the Swedish cancer register.

We selected data from the Molecular Taxonomy of Breast Cancer International Consortium (METABRIC cohort, *n* = 1980 patients), EGAC00001000484 [34], and the recently published, population-based multicenter Sweden Cancerome Analysis Network—Breast Initiative cohort (SCAN-B, *n* = 3678 patients), GSE60788 [35]. The METABRIC data were based on microarray analysis, while the SCAN-B cohort used RNA sequencing (RNA-seq) for transcriptomics. Information about the cohorts is shown in Table 1.

Tumor cytosol specimens

Tumor tissues were obtained from the fresh-frozen tumor bank at the Sahlgrenska University Hospital Oncology Lab (Gothenburg, Sweden). Tissue lysates were prepared by glycerol-sucrose extraction (0.5 M in 3 mM MgCl_2_ and 20 mM 4-(2-hyroxyethyl)-1- piperazineetha-nesulfonic acid, pH 7.4) [36]. The tumor lysates were stored at −80 °C before use.

Hormone receptors analysis

Estrogen receptor (ER) and progesterone receptor (PR) were precipitated using monoclonal antibodies coated on polystyrene beads. Quantification of ER and PR (progesterone receptor) was done following the routines at the Laboratory of Clinical Chemistry, Sahlgrenska University Hospital, using an EIA kit (Abbott, Chicago, IL, USA).

Protein survivin levels analysis

Survivin levels were measured in tumor lysates (diluted 1:20 in 1% bovine serum albumin (BSA, Sigma, St. Louis, MO, USA) in phosphate buffer saline), using the Human Total Survivin DuoSet IC ELISA (DYC647E, R&D Systems, Minneapolis, MN, USA), a sandwich enzyme-linked immunoassay consisting of a pair of matched antibodies and recombinant standard, as described [37].

The total protein concentration was quantified in BC tissue lysates diluted 1:10 in physiological saline solution, using the Pierce BCA Protein Assay kit (ThermoFisher, Rockford, IL, USA), and was used to normalize the measurements of survivin levels. The ELISA and BCA protein assay were measured with a Spectramax 340 Microplate Reader (Molecular Devices, Sunnyvale, CA, USA). Survivin concentration was normalized by the total protein content in BC lysate and expressed in pg[survivin]/g[total protein]. Samples with survivin level below 100 pg/g were considered survivin negative (Survivin0), the values of 100–1000 pg/g were considered survivin low (SurvivinLO), and levels above 1000 pg/g were survivin high (SurvivinHI).

Analysis

Transcriptomic cohorts information was extracted using the CBIOportal (http://www.cbioportal.org/, accessed on 12 February 2020) [38]], NCBI GEO (https://www.ncbi.nlm.nih.gov/gds, accessed on 12 February 2020), and BC data miner [39,40]. Gene expression levels in transcriptomic cohorts were based on MAS5-normalized and log2 transformed values for microarray and cufflinks2 FPKM normalized log2 transformed values for RNAseq information. For the cross-cohort comparison of survival, information of the five-year data was extracted for all cohorts. The time to endpoint was selected for the overall survival analysis for the METABRIC, SCAN-B, and VGR-BC cohorts, and the disease-specific survival for VGR-BC cohort. BC survival rates were defined as the time from initial diagnosis to BC-related death for the disease-specific survival, and the time from initial diagnosis to death from any cause for the overall survival. Survival analysis was performed using Kaplan-Meier curves with log-rank test and univariate Cox proportional hazard model, using the R/BioConductor (v.3.6.0, packages survival (version 2.44-1.1) [41,42], and survminer (version 0.4.6), (http://www.sthda.com/english/rpkgs/survminer/, accessed on 12 February 2020). For the statistical tests, an additional JASP statistic package (https://jasp-stats.org/, v.0.13, accessed on 12 February 2020) was used.

BIRC5/survivin expression levels were compared for each cohort between the groups with different BC characteristics, such as lymph node metastases status, ER/PR status according to immunohistological annotation of samples, age at diagnosis, tumor size, and histological grading [4] and Sorlie’s molecular subtype classification [43] for the METABRIC and SCAN-B cohorts. Datasets were checked for normality using the Shapiro-Wilk test. For pairwise group comparisons, the *t*-test in Welch modification, which allowed unequal variances, was used. For the VGR-BC, non-parametric Mann-Whitney test was applied. For multiple group comparisons, the initial pairwise Welch test was applied, followed by the Dunnett-Tukey-Kramer’s test, which permitted the two-by-two comparisons and showed if the significant difference existed. Additional one-way ANOVA test was applied for the *p*-value estimation between multiple groups. Samples with missing values were excluded. *p*-value below 0.05 was selected as the levels of significant difference for all tests

## 3. Results

### 3.1. Intercohort Analysis of BIRC5 Level in Breast Cancer Patients

#### 3.1.1. High Expression of BIRC5/Survivin in Primary BC Tissue Is Associated with Poor Survival in Three Independent BC Cohorts

Survivin protein levels, product of the BIRC5 gene, were measured in 845 BC tissue lysates of the VGR-BC cohort. Among those BC samples, 66 (7.8%) had survivin levels under the detection limit. The long-term survival analysis was performed for the periods of 5 and 27 years to compare these patients with the groups annotated as low or high survivin levels (Figure 1A,B, Table 2). The Mantel-Cox analysis revealed that the group with no detectable survivin had a significantly better survival outcome, as compared to the groups with high and low BIRC5/survivin levels (Figure 1A,B, Table 2). The results were consistent for the comparison of DSS (disease-specific survival) and OS (overall survival), and also after the periods of 5 and 27 years (Table 2). The three groups showed statistically significant difference in the survival outcome at 5 years (OS *p* = 1.3 × 10^−5^) and 27 years (OS *p* = 0.002, DSS *p* = 0.036). Pairwise Bonferroni corrected differences reached statistical significance for DSS “0” vs. low (5 and 27 years), DSS “0” vs. high (27 years), and OS “0” vs. low (5 years) (Table 2). To investigate if patients with no measurable BIRC5/survivin were different from all others, we compared the “0” patients with the combined “high + low” BIRC5/survivin groups. We demonstrated that the group with no BIRC5/survivin had significantly better survival (both DSS and OS, Table 2).

To compare clinical association of BIRC5/survivin protein levels and the expression of its mRNA, we took advantage of the mRNA levels of samples for the independent well-characterized BC cohorts, METABRIC [34], and SCAN-B (Table 1, Figure 1C) [35]. The OS analysis in these independent cohorts was performed on the total dataset split according to the median BIRC5/survivin level. It demonstrated a significantly worse 5-year survival prognosis in association with the higher BIRC5/survivin levels within each cohort (hazard ratio (HR) 1.12 [1.03–1.57], *p* = 0.028 for VGR-BC, HR 1.62 [1.33–1.98], *p* < 0.0001 for SCAN-B, HR 2.15 [1.78–2.60], *p* < 0.0001 for METABRIC) (Figure 1C). These consistent results were obtained in the cohorts with different OS, and could be attributed to the changes in cancer treatment in recent decades. The cross-cohort variations in the tumor size, hormonal receptors, and nodal status and patient age at diagnosis should not be overlooked.

#### 3.1.2. High BIRC5/Survivin Expression in BC Tissue Indicates Poor Survival Independent of the Nodal and Hormonal Receptor Status of BC

Next, we wanted to study if the high expression of BIRC5/survivin was associated with other survival prognostic factors. To analyze this, the cohorts were split into subgroups according to clinically apparent prognostic parameters, the presence of metastasis into lymph nodes, and the expression of hormonal receptors in BC tissue (Figure 2).

Survival analysis of subgroups with the nodal positive (N+) and nodal negative (N−) status revealed that high BIRC5/survivin levels were associated with poor 5 years survival both in the N+ and in N− groups, which supported the notion of BIRC5/survivin being a prognosis marker independent of and regardless of the nodal status. For the VGR-BC cohort, the difference between the N+ and N− groups reached no statistical significance, due to the smaller sample size (351 N+ and 236 N− samples) of the VGR-BC cohort. While for the METABRIC and SCAN-B, high BIRC5/survivin mRNA levels were significantly associated with worse survival in the N+ and in N− subgroups (N−, HR 1.45 [1.09–1.93], *p* 0.0104; N+, HR 1.84 [1.35–2.50]; *p* < 0.0001 for SCAN-B. N−, HR 1.80 [1.30–2.50], *p* = 0.0005, N+, HR 1.78 [1.42–2.25], *p* < 0.0001 for METABRIC) (Figure 2A). Both, the SCAN-B and METABRIC cohorts showed the increased expression of the BIRC5 transcript in tumors of BC patients with nodal metastasis (Welch *p*-values < 0.0001 and 0.0002, respectively) (Figure 3A).

The survival analysis of patients differing in expression of ER showed that high BIRC5/survivin was associated with poor survival in both the ER-positive and ER-negative groups. In the SCAN-B cohort, high BIRC5 levels associated with significantly lower 5-year survival prognosis (ER-, HR 1.95 [1.06–3.31], *p* = 0.035; ER+, HR 1.63 [1.30–2.04], *p* < 0.0001). In the METABRIC cohorts, the difference in 5-year survival between the high and low BIRC5/survivin was significant for ER+ group (ER+, HR 1.86 [1.45–2.87], *p* < 0.0001) (Figure 2B), while within the ER-negative group with smaller number of samples, no statistically significant difference was reached. For VGR-BC, due to the smaller dataset, heterogeneous OS outcome analysis did not achieve statistical significance, but the tendencies were similar to that showed in Figure 2B. The OS and DSS analysis supported the notion of poor prognosis for high BIRC5/survivin in the VGR-BC ER-negative groups (Table 2).

BIRC5/survivin was expressed at higher levels in the ER-negative tumors in all three cohorts (Figure 3B, VGR-BC *p* = 0.021, SCAN-B, and METABRIC *p* < 0.0001). All three cohorts demonstrated similar distribution of BIRC5/survivin levels without signs of population heterogeneity. Similar differences in BIRC5/survivin levels were detected in the PR-negative BC samples, as compared to PR+ (Figure 3E, SCAN-B, *p* < 0.0001; VGR-BC, *p* = 0.027). While the hormonal receptor negative (HRN) VGR-BC samples and triple negative (TNBC) SCAN-B samples both showed higher BIRC5 level (VGR-BC, *p* = 0.057; SCAN-B, *p* < 0.0001) (Figure 3F).

#### 3.1.3. BIRC5/Survivin Expression Is Related to the Specific Molecular Subtype and Increasing Histological Grading of BC

Analysis of BIRC5/survivin mRNA expression levels with respect to the histological grading (according to Scarff-Bloom-Richardson, SBR [3]) of the tumors demonstrated that BIRC5 mRNA was step-wise and significantly increased with each histological grade. Indeed, BIRC5/survivin mRNA levels grew according to increasing histological grade in both cohorts (Dunnett-Tukey-Kramer’s two-by-two test passed significance level, one-way ANOVA *p* < 0.0001 for both cohorts) (Figure 3C).

Analysis of BIRC5/survivin mRNA in the BC molecular subtypes [6,43] showed a high similarity for the METABRIC (microarray data) and SCAN-B (RNA-seq data) cohorts. The normal breast-like BC subtype had the lowest BIRC5 mRNA expression, while the highest levels recognized luminal B and basal-like BC subtypes. For both cohorts, one-way ANOVA with Dunnett-Tukey-Kramer’s post-hoc test supported significant differences between the molecular subtypes within the METABRIC and the SCAN-B cohorts (Figure 3D).

## 4. Discussion

In the present study, we applied the analysis of BIRC5/survivin expression in the primary BC tissue in the material of three independent BC cohorts, and demonstrated that high mRNA and protein BIRC5/survivin levels were associated with low probability of survival. Two of the cohorts, VGR-BC and METABRIC, were recruited from early 70s, while the third, SCAN-B cohort started patient recruitment from the beginning of 2010s. This meant that we compared clinical outcomes in the datasets, which enrolled and followed BC patients separated by 2–3–4 decades. The survival curves of these different cohorts justify the current progress in BC diagnosis and treatment [44,45]. Despite being diagnosed in different decades and most likely receiving different treatment regimens, the patients with high BIRC5/survivin tended to show an adverse survival outcome. In contrary, we observed a significantly better five-year survival probability for the BC patients with low BIRC5/survivin expression.

The analysis of the relation between the BIRC5/survivin expression and most established clinical prognostic and predictive indicators, nodal status at diagnosis and hormone receptor expression, demonstrated that high BIRC5/survivin expression levels were characteristic of distinct tumor subsets. Higher levels of BIRC5/survivin were associated with worse prognosis both for the ER-positive and ER-negative BC, which was consistently demonstrated in the protein-based and mRNA-based analysis of our independent cohorts. However, the ER-negative BC tumors were generally associated with significantly higher BIRC5/survivin expression. This matched the previously reported findings [46]. For the METABRIC cohort, high BIRC5/survivin expression in the ER-positive BC was associated with the poor 5-year overall survival. This survival difference was not significant in the smaller subset of the ER-negative patients. Similar to the VGR-BC and SCAN-B cohorts, the absolute BIRC5/survivin levels in the METABRIC cohort were still significantly higher in the ER-negative BC, as compared to those ER-positive. The long-term 27-years survival data were available for VGR-BC and revealed that zero BIRC5/survivin protein levels in the tumor samples carried the important message for our understanding of the prognostic survival differences. The patients with zero-survivin status demonstrated the best disease-specific survival, as compared to those with low, but detectable BIRC5/survivin expression (Figure 1A,B, Table 2). Moreover, these patients showed better OS and DSS outcomes, both at 5- and 27-years, in comparison to all others combined (Table 2).

We observed that high expression of BIRC5/survivin was associated with specific molecular subtypes of BC. The analysis demonstrated strikingly similar expression patterns between the molecular subtypes within the SCAN-B and METABRIC cohorts. Higher BIRC5/survivin levels were attributed to the basal-like and luminal B subtypes of BC. Such increased expression of BIRC5/survivin emerged to be one of the major markers supporting the distinction between the luminal A and luminal B BC subtypes [47,48]. Interestingly, the information about molecular BC subtypes was available for the METABRIC and SCAN-B cohorts and clearly illustrated that high expression of BIRC5/survivin complemented the survival probability curves within the ER-positive patients, which are mostly the luminal B molecular subtype, and ER-negative, which are mostly the basal-like subtype. These subtypes are recognized by aggressive tumor growth and are prone to develop primary or acquired therapy resistance with high frequency of recurrence [49,50]. These observations suggest that measurement of BIRC5/survivin could be concordant with the detailed immune gene expression profile for patient’s outcome in the luminal BC [51,52,53]. Additionally, high BIRC5/survivin levels were identified as a sensitive individual indicator for the locally advanced BC not responding to neoadjuvant chemotherapy [54,55,56,57]. This is yet an additional argument in favor of measurement of BIRC5/survivin protein or mRNA levels in biopsies, or surgically removed tumors. Information of BIRC5/survivin expression could assist in recognition of therapy-resistant BC. BIRC5/survivin is a potential cancer drug target, which requires careful selection of patients to further address this hypothesis. Moreover, the association between low BIRC5/survivin levels and better survival prognosis is appealing for different clinical decisions, when a step-down strategy might be applied to avoid or reduce unneeded cytotoxicity of chemotherapy.

In the total material of this study, we included 6503 BC patients from three independent BC cohorts. It was almost 2-times bigger as compared to only the comprehensive meta-analysis of 3259 BC patients reported in 2014 [24]. Importantly, the quantitative analysis of BIRC5/survivin levels was possible in all three cohorts and the results were reproducible both for the measurement of BIRC5/survivin protein (VGR-BC cohort) and transcript (METABRIC and SCAN-B cohort) levels in BC tissue. Previous studies employed IHC, which allowed visual assessment of BIRC5/survivin protein localization in the cells [24], but provided no quantitative measures for comparison between samples and the different patient cohorts. Several recent reports presented divergent results in an attempt to use the cellular localization of BIRC5/survivin for prediction of BC recurrence [25,58,59]. We demonstrated that BIRC5/survivin analysis on the mRNA or protein levels was sufficiently sensitive to predict clinical outcome, disregarding its cellular localization.

The intracellular mechanisms of BIRC5/survivin involvement in BC progression are still unclear. The results of our study suggest that BIRC5/survivin expression is independent of the hormone receptor status of BC and is unlikely and not solely regulated through the ER signaling. Experimental evidence supports the functional STAT3-survivin and Notch-survivin connection in BC cells that lack hormone receptors [60,61]. Much attention is currently given to the proliferation-related mechanism of BC due to the functional specificity of BIRC5/survivin in activating Aurora B kinase within the chromosomal passenger complex, and its co-localization with other tumor proliferation genes MKI67, CCNB1, and AURKA. We observed that BIRC5/survivin expression was significantly associated with the higher histological grade of BC, which reflected a degree of BC deviation from normal tissue. The direct positive association of BIRC5/survivin and histological grading seems natural keeping in mind that this protein is principally expressed in the G2 and the mitotic phases of cell division. In concordance with this, a difference in tumor size was not supported by analysis of BIRC5/survivin in the METABRIC and SCAN-B cohorts. Several recent studies presented experimental evidence for the role of BIRC5/survivin in cancer cell motility, including the melanoma model that revealed the α5-integrin pathway [62], cervical carcinoma [63], and others. For BC, a connection between BIRC5/survivin expression and development of the lymph node metastases varied among studies [60,64,65]. We showed a significant association of high BIRC5/survivin expression with poor survival. This finding was independent of the nodal status at BC diagnosis and provided important clinical information for the patients with and without the lymph node metastasis. This, however, did not exclude a functional connection between the increased BIRC5/survivin expression and the capability of BC for metastasis. Additionally, the high expression of BIRC5/survivin could represent pro-inflammatory and immune-associated mechanisms that are active within all BC subtypes.

The recent attention to the role of tumor-infiltrating leukocytes and immune gene signatures in BC progression and survival, points to the existing differences between the known subtypes, and ask for further improvement of this molecular-based classification [66]. Gene expression profiling tests are rapidly coming into clinical use today with the ultimate goal to provide crucial prognostic information on treatment response and the assessment of recurrence risk or death of BC. The very recent health economic analyses [67,68] imply that the commercially available gene signature tests OncotypeDX, Prosigna/PAM50, and EndoPredict are cost effective and superior to the current practice of providing important prognostic information for certain patient subgroups [69,70]. BIRC5/survivin is included in most clinically promising gene profiling tests of BC, which clearly reflects its high predictive and prognostic value for this heterogeneous disease. According to the clinical trials data, the multi gene profiling tests demonstrated their importance for clinical decisions, while most studies are still ongoing and the results are not yet available. The BIRC5 gene was included in some studies as a therapy target or as one of the markers, according to the IHC nuclear staining or a component of the multi gene panels. However, the single-gene quantitative analysis of BIRC5 was not thoroughly investigated for its clinical significance.

## 5. Conclusions

Taken together, the results of our study indicate BIRC5/survivin to be a sensitive single-gene maker of survival probability in BC acting independently of the ER and the nodal status of patients. Quantitative analysis of BIRC5/survivin expression at the mRNA or the protein level, needs to be considered when making treatment decisions in BC patients.

## Figures and Tables

**Figure 1 cancers-13-02209-f001:**
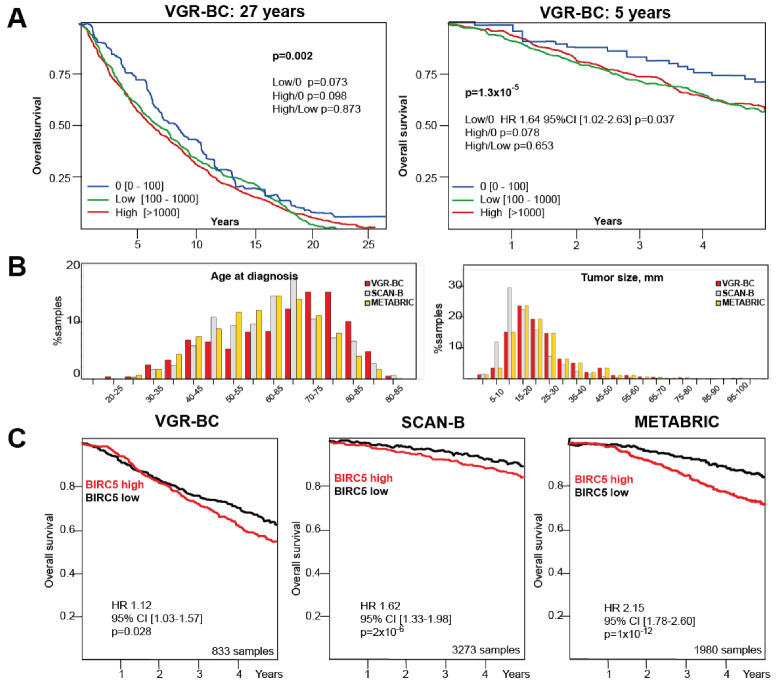
BIRC5/survivin levels contribute to survival probability in three independent breast cancer cohorts of different data and clinical characteristics. (**A**) Kaplan-Meier curves depict 27 and 5 years of overall survival for the VGR-BC cohort, split by the BIRC5/survivin levels into “0”, below the threshold (*n* = 66); “Low” (100–1000 pg/1 g total protein, *n* = 536) and ”High” ( >1000 pg/1 g total protein, *n* = 234) BIRC5/survivin levels. Mantel-Cox log rank test was used for pairwise comparisons. (**B**) Comparative analysis of VGR-BC, SCAN-B, and METABRIC cohorts. Histograms present the age distributions at diagnosis and tumor size within VGR-BC, SCAN-B, and METABRIC cohorts. Percent of total samples size was independently calculated for each cohort. (**C**) Kaplan-Meier curves depict 5-year overall survival probability for patients of the VGR-BC, SCAN-B, and METABRIC cohorts. Survival curves shown for total datasets, split to the BIRC5/survivin median level was done based on the protein levels in the tumor lysates in the VGR-BC, and mRNA levels in SCAN-B and METABRIC. Hazard ratio was estimated separately for the high- and low-level patient groups.

**Figure 2 cancers-13-02209-f002:**
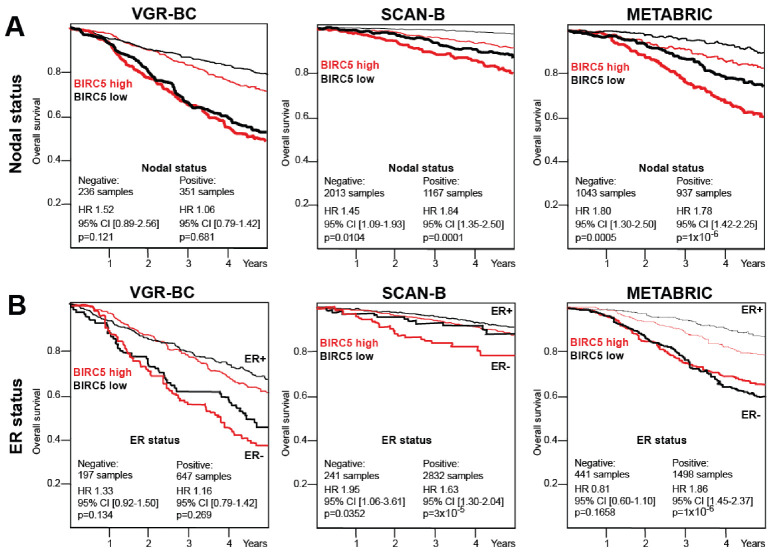
High levels of BIRC5/survivin are associated with poor survival prognosis independent of nodal and ER status. (**A**) Kaplan-Meier curves depict 5-year overall survival probability for patients of the VGR-BC, SCAN-B, and METABRIC cohorts. Data were split by BIRC5/survivin median levels of the protein in tumor lysates in VGR-BC and of the mRNA levels in SCAN-B and METABRIC. Graphs are shown for patients with (nodal positive, thick lines) and without (nodal negative, thin lines) BC metastases into lymph nodes. (**B**) Kaplan-Meier curves depict 5-year overall survival probability for the ER-positive (ER+) and ER-negative (ER−) patients within the VGR-BC, SCAN-B, and METABRIC cohort. Data were split by BIRC5/survivin median levels of the protein in tumor lysates in VGR-BC and of the mRNA levels in SCAN-B and METABRIC. High BIRC5/survivin level survival curves are shown in red and low in black. Hazard ratio was estimated separately for high- and low-level patient groups.

**Figure 3 cancers-13-02209-f003:**
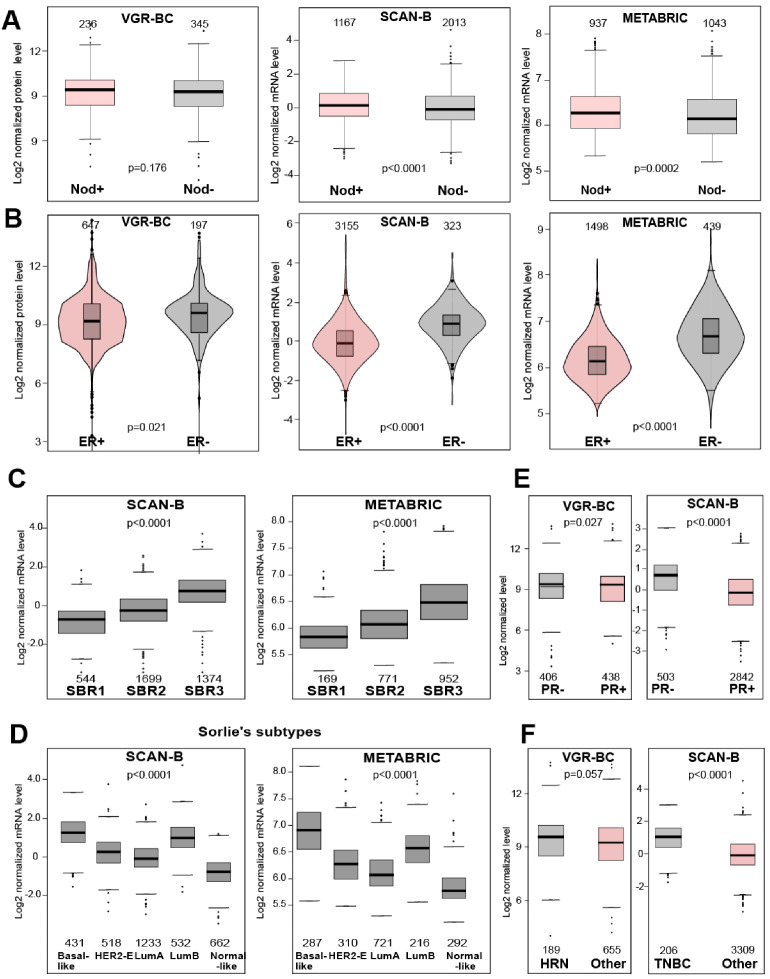
BIRC5/survivin level variation across BC subtypes and clinical characteristics. (**A**) Boxplots show log2 normalized BIRC5/survivin levels in nodal-positive and nodal-negative tumors. *p*-values are obtained by the Welch and Mann-Whitney tests. Numbers indicate the sample size of the group. (**B**) Violin plots show the log2 normalized BIRC5/survivin levels in ER-positive and ER-negative tumors in the VGR-BC (protein data) and SCAN-B (RNA-seq data) cohorts. *p*-values are obtained by the Welch (SCAN-B and METABRIC) and Mann-Whitney (VGR-BC) tests. Numbers indicate the sample size of the group. (**C**) Boxplots show log2 normalized BIRC5/survivin mRNA levels in tumors with histological grading (according to Scarff-Bloom-Richardson grading, SBR) for the SCAN-B and METABRIC cohorts. *p*-values are obtained by the ANOVA test. (**D**) Boxplots show the log2 normalized BIRC5/survivin mRNA levels in the samples from the SCAN-B and METABRIC cohorts classified to different molecular subtypes (Sorlie’s). *p*-values are obtained by the ANOVA test. Numbers indicate the sample size of the group. (**E**) Boxplots show log2 normalized BIRC5/survivin levels in the PR-positive and PR-negative tumors. *p*-values are obtained by the Welch (SCAN-B) and Mann-Whitney (VGR-BC) tests. Numbers indicate the sample size of the group. (**F**) Boxplots show the log2 normalized BIRC5/survivin levels in the HRN and TNBC tumors. *p*-values are obtained by the Welch (SCAN-B) and Mann-Whitney (VGR-BC) tests. Numbers indicate the sample size of the group.

**Table 1 cancers-13-02209-t001:** Clinical characteristics of the breast cancer cohorts.

KERRYPNX	VGR-BC	SCAN-B	METABRIC
Total number	845	3678	1980
Years	1989–1995	2010–2015	1963–2005
Data type	Protein, ELISA normalized to total protein	mRNA, RNAseq, polyA-based cDNA library, paired-end, Illumina HiSeq 2000	mRNA, microarray, Illumina, GPL6947, HumanHT-12 V3.0
Age at diagnosis, years	65.64 ± 14.83	62.74 ± 13.15	60.42 ± 13.03
(mean ± SD, *p*-value vs. VGR-BC)		*p* = 1 × 10^−8^	*p* = 1 × 10^−20^
Tumor size, mm	30.49 ± 17.12	19.91 ± 12.18	26.22 ± 15.34
(mean ± SD, *p*-value vs. VGR-BC)		*p* = 5 × 10^−93^	*p* = 7 × 10^−11^
ER	76.69%	90.71%	77.34%
positive	*n* = 845	*n* = 3155	*n* = 1498
PR	51.83%	84.96%	52.53%
positive	*n* = 845	*n* = 3345	*n* = 1980
HER2	Not available	14.23%	12.47%
positive		*n* = 3556	*n* = 1980
LN metastases	59.38%	36.70%	47.32%
positive	*n* = 581	*n* = 3180	*n* = 1980

**Table 2 cancers-13-02209-t002:** Overall and disease-specific survival analysis for the VGR-BC cohort.

A. BIRC5/Survivin Protein Level [pg/g], Annotated Groups			
0, *n* = 66	Range [0–100], 20 ± 31		
Low, *n* = 544	Range [100–1000], 520 ± 247		
High, *n* = 235	Range [1000–15,155], 2186 ± 137		
Survival probability		27 years	5 years
DSS	Overall	*p* = 0.036	*p* = 0.062
	High vs. 0	HR 1.57	*p* = 0.082
		95%CI [1.07–2.33], *p* = 0.023	
	Low vs. 0	HR 1.62	HR 1.92
		95%CI [1.12–2.32], *p* = 0.011	95%CI [1.09–3.33], *p* = 0.022
	High vs. Low	*p* = 0.804	*p* = 0.365
OS	Overall	*p* = 0.002	*p* = 1.3 × 10^−5^
	High vs. 0	*p* = 0.098	*p* = 0.078
	Low vs. 0	*p* = 0.073	HR 1.64
			95%CI [1.02–3.22], *p* = 0.037
	High vs. Low	*p* = 0.873	*p* = 0.653
Low + High, *n* = 779	Range [100–15,151], 1023 ± 1400		
DSS	Low + High vs. 0	HR 1.60	HR 1.85
		95%CI [1.11–2.31], *p* = 0.01	95%CI [1.05–3.22], *p* = 0.033
OS	Low + High vs. 0	*p* = 0.065	HR 1.61
			95%CI [1.01–2.56], *p* = 0.043
**B. BIRC5/Survivin Protein Level [pg/g], Median Split**			
Low, *n* = 422	Range [0–630.06], 312 ± 181		
High, *n* = 423	Range [632.85–15,155], 1574 ± 1711		
Survival probability		27 years	5 years
DSS	High vs. Low	HR 1.31	HR 1.29
		95%CI [1.10–1.56], *p* = 0.002	95%CI [1.01–1.64], *p* = 0.038
DSS, ER-positive	High vs. Low	*p* = 0.260	*p* = 0.823
DSS, ER-negative	High vs. Low	HR 1.76	HR 1.59
		95%CI [1.24–2.49], *p* = 0.0013	95%CI [1.05–2.39], *p* = 0.027
OS	High vs. Low	HR 1.38	HR 1.12
		95%CI [1.03–1.85], *p* = 0.033	95%CI [1.03–1.57], *p* = 0.028
OS, ER-positive	High vs. Low	*p* = 0.367	*p* = 0.269
OS, ER-negative	High vs. Low	HR 1.38	*p* = 0.134
		95%CI [1.03–1.85], *p* = 0.031	

Survival was calculated by Mantel-Cox statistics. Protein levels in groups are indicated as mean ± std.

## Data Availability

Data for VGR-BC cohort available upon reasonable request. Public data sources listed in the references (EGAC00001000484 for METABRIC cohort and GSE60788 for the SCAN-B cohort.

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
