# Peer review of "Prognostic Significance of BIRC5/Survivin in Breast Cancer: Results from Three Independent Cohorts"

_cancers, 2021, doi:10.3390/cancers13092209_

Round 1
Reviewer 1 Report
The authors report a complete study on the prognostic significance of BIRC5/Survivin in breast cancer using in three three independent breast cancer cohorts (VGR-BC, SCAN B, METABRIC), reaching a total of 6503 patients. Moreover, they provide data on accurate quantitative measurement of protein level of BIRC5/Survivin in tumor lysates in the VGR-BC cohort. They show that high BIRC5/survivin levels (measured as protein or mRNA) are associated with poor survival probability of BC patients. The paper is well-written and data are of relevance in the specific field. This referee has only some minor points:
Line 76: “chromatin passenger complex” should be “chromosome passenger complex”
Line 100 “pomissing” should be “promising”
Line 216-218: “Overall survival outcome for three groups was significant for 5 years (OS p=0.000013) and 27 years (OS p=0.002, DSS p=0.036).” The term “overall” in this sentence (and in table 2) is rather confusing (the same adjective is used for simultaneous comparison of the three groups (0, high and low) (in contrast to pairwise comparison) and for a type of survival analysis (overall survival). It is possible to modify this sentence in the following way: “The three groups showed statistically significant difference in the survival outcome at 5 years (OS p=0.000013) and 27 years (OS p=0.002, DSS p=0.036)”.
Table 2 “pg/gl” should be “pg/g”.
Line 220: “were differed” should be “were different”.
Line 239-240: “These consistent results were obtained in the cohorts differed in the OS….”. Please, reformulate this sentence.
Figure 1 and 2: Although it is correctly stated in “material and methods” and in “discussion” that BIRC5/Survivin was measured at the protein level in the VGR-BC cohort and at mRNA level in the SCAN B and METABRIC cohort, it is better to clarify this point also in the legend of figure 1c and figure 2a and b.
Author Response
The authors would like to express their sincere appreciation of the work done by reviewers on our manuscript. We have now introduced the changes in the text of the manuscript suggested by the reviewers.
Please, find below our point-by-point revision of the text.
Reviewer 1:
Thank you for keeping your keen eye on these important details. We have now corrected and re-phrased the text as you suggest. The changes are marked yellow through out the text.
Reviewer 2 Report
Do the authors focused on the patient age as a factor associated with survivin expression? IS there any differencie in expression and prognostic value in women under the age of 40 ?
The article is of high quality. I recommend it for the publication.
Author Response
The authors would like to express their sincere appreciation of the work done by reviewers on our manuscript. We have now introduced the changes in the text of the manuscript suggested by the reviewers.
Breast cancer in young patients is an important clinical issue. Age distribution at diagnosis is depicted in figure 1B and covers all 3 BC cohorts. As you may see in the figure 1B, only about 10% of all patients were of age below 45y, which limits sensitivity of prediction. Integrated meta-analysis of larger number of cohorts aiming to study this specific scientific question will probably provide an answer to the question.